# Protocol for hypofractionated adaptive radiotherapy to the bladder within a multicentre phase II randomised trial: radiotherapy planning and delivery guidance

Shaista Hafeez [iD] ,[1,2] Emma Patel,[3] Amanda Webster,[3] Karole Warren-Oseni,[1,2] Vibeke Hansen,[4] Helen McNair,[1,2] Elizabeth Miles,[3] Rebecca Lewis,[5] Emma Hall,[5] Robert Huddart[1,2]

For numbered affiliations see end of article.

**Correspondence to**
Dr Shaista Hafeez;
shaista.hafeez@icr.ac.uk

## ABSTRACT

**Introduction** Patients with muscle invasive bladder cancer (MIBC) who are unfit and unsuitable for standard radical treatment with cystectomy or daily radiotherapy present a large unmet clinical need. Untreated, they suffer high cancer specific mortality and risk significant disease-related local symptoms. Hypofractionated radiotherapy (delivering higher doses in fewer fractions/visits) is a potential treatment solution but could be compromised by the mobile nature of the bladder, resulting in target misses in a significant proportion of fractions. Adaptive 'plan of the day' image-guided radiotherapy delivery may improve the precision and accuracy of treatment. We aim to demonstrate within a randomised multicentre phase II trial feasibility of plan of the day hypofractionated bladder radiotherapy delivery with acceptable rates of toxicity.

**Methods and analysis** Patients with T2–T4aN0M0 MIBC receiving 36 Gy in 6-weekly fractions are randomised (1:1) between treatment delivered using a single-standard plan or adaptive radiotherapy using a library of three plans (small, medium and large). A cone beam CT taken prior to each treatment is used to visualise the anatomy and select the most appropriate plan depending on the bladder shape and size. A comprehensive radiotherapy quality assurance programme has been instituted to ensure standardisation of radiotherapy planning and delivery. The primary endpoint is to exclude ≥30% acute grade ≥3 non-genitourinary toxicity at 3 months for adaptive radiotherapy in patients who received ≥1 fraction (p0=0.7, p1=0.9, α=0.05, β=0.2). Secondary endpoints include local disease control, symptom control, late toxicity, overall survival, patient-reported outcomes and proportion of fractions benefiting from adaptive planning. Target recruitment is 62 patients.

**Ethics and dissemination** The trial is approved by the London-Surrey Borders Research Ethics Committee (13/LO/1350). The results will be disseminated via peer-reviewed scientific journals, conference presentations and submission to regulatory authorities.

**Trial registration number** NCT01810757.

### Strengths and limitations of this study

► This is a phase II national multicentre randomised control trial evaluating innovation in radiotherapy technology (strength).

► The trial has a non-comparative single-stage design (limitation).

► Detailed guidance and training for this novel radiotherapy technique are provided to ensure standardisation across multiple participating centres (strength).

► A robust pretrial and on trial radiotherapy quality assurance programme is in place to ensure standardisation of a trial technique (strength).

► Primary endpoint focus is based on determining early effectiveness of this approach as measured by acute non-genitourinary grade three toxicity scoring (strength).

## INTRODUCTION

Standard radical management of muscle invasive bladder cancer (MIBC) involves either radical cystectomy or a course of daily radiotherapy delivered with radiosensitisation over 4–7 weeks.[1–5] Given the aetiological association of bladder cancer with smoking, cardiovascular and respiratory comorbidities are common.[6 7] Undertreatment and poor access to effective treatment are particularly evident in older patient groups who have the highest risk of cancer related morbidity and death from initially curable bladder cancer.[8]

Hypofractionated radiotherapy (delivering higher doses in fewer fractions/visits) may provide a potential treatment solution for these patients. The only multicentre randomised control trial of hypofractionated bladder radiotherapy investigated two schedules of relatively low biological effectiveness;

**BMJ**

35 Gy in 10 fractions over 2 weeks and 21 Gy in 3 fractions over 1 week.[9] Both treatment groups achieved similar symptom control with no significant difference in efficacy or toxicity evident between different radiotherapy schedules. Despite the palliative treatment intent, approximately 20% of patients achieved survival beyond 24 months.[9] Given the presumed dose–response relationship of MIBC to radiotherapy, a higher biological effective dose would be expected to improve local disease and symptom control further.[10]

A number of small single-centre studies using the higher biological dose of 30–36 Gy in 6 Gy per fraction suggest acceptable acute and late toxicity with local control achieved in over of 60% patients at 3 months.[11–13] Prospective multicentre assessment of this radiotherapy schedule has not yet been performed.

Reliably targeting the bladder for radiotherapy is challenging. It is a relatively mobile structure subject to marked shape and volume change during a course of radiotherapy.[14–16] This has meant that historically bladder cancer radiotherapy has been delivered with some element of geographical miss (up to 57% of fractions) even when large safety margins of upto 1.5 cm are applied to create the planning target volume (PTV).[17] The expected consequence of dose intended for the target hitting adjacent normal structures is reduced tumor control and increased treatment related toxicity. Larger safety margins would more reliably encompass the bladder target variation but would further increase the normal tissue exposed to radiation dose, so increase side effects from treatment.

Volumetric soft-tissue imaging made possible by cone beam CT (CBCT) technology integrated on current generation linear accelerators allows a three-dimensional image to be acquired immediately prior to treatment. This informs positional adjustment to optimise target coverage by the radiotherapy plan. It also has enabled 'plan of the day' solution. Rather than a single plan available for treatment, a library of plans can be created to cover the range of expected filling and positional variation of the bladder. Acquiring CBCT just prior to treatment allows visualisation of the soft tissue so that a plan which best covers the bladder target with least normal tissue irradiation can be selected for treatment that day.[17]

In a single-centre non-randomised phase II study, we demonstrated feasibility of the plan of the day approach using library of three plans in a patient population with MIBC unfit for radical treatment.[18] Target coverage was maintained with reduction in dose to normal tissue irradiation compared with single standard plan.[19] The multicentre randomised phase II study of HYpofractionated Bladder Radiotherapy with or without Image guided aDaptive planning (HYBRID) seeks to examine whether this treatment approach can be consistently and safely delivered across multiple National Health Service (NHS) centres.

Below, we describe the HYBRID trial protocol with particular emphasis on the radiotherapy procedural aspects, including preparatory imaging, treatment planning, delivery and evaluation, with the aim of providing comprehensive description of the radiotherapy implemented for the study.

## Hypothesis

Adaptive radiotherapy techniques can be delivered at multiple centres and result in acceptable levels of acute non-genitourinary (GU) side effects experienced by patients with MIBC unsuitable for radical daily radiotherapy or cystectomy.

## MATERIALS AND ANALYSIS
### Study design

HYBRID is a non-blinded multicentre non-comparative randomised control phase II trial conducted in accordance with the Research Governance Framework for Health and Social Care and principles of Good Clinical Practice. The trial is sponsored by The Institute of Cancer Research, registered on the ClinicalTrials.gov database and is included in the National Institute for Health Research (NIHR) Clinical Research Network portfolio. The final ethics approved version of HYBRID trial protocol is provided in the online supplementary file 1.

All patients are planned to receive a total dose of 36 Gy in 6-weekly fractions randomised (1:1) between treatment delivered using a single standard plan (control) or adaptive radiotherapy using a library of plans. Randomisation takes place centrally by the trials unit (Clinical Trials and Statistics Unit at The Institute of Cancer Research (ICR-CTSU)) within a maximum of 6 weeks prior to the planned radiotherapy start date.

The primary endpoint is to evaluate acute non-GU grade 3 or greater toxicity as assessed using Common Terminology Criteria for Adverse Events (CTCAE V.4). The secondary endpoints are to assess local disease control at 3 months, control rate of presenting symptoms as measured by CTCAE V.4, patient-reported outcomes as measured by Inflammatory Bowel Disease Questionnaire, King's Health Questionnaire and EQ5D, late toxicity as measured by CTCAE V.4 and the Radiation Therapy Oncology Group (RTOG) late radiation morbidity scoring schema, time to local disease progression, overall survival and proportion of fractions benefiting from adaptive planning.

The trial has a number of exploratory secondary endpoints related to the appropriate identification of plan selection, target coverage and concordance between clinical and patient-reported outcomes.

Figure 1 shows the trial schema and overview of follow-up. Table 1 provides summary of the scheduled prerandomisation, on treatment, and post-treatment assessments.

### Participants and eligibility

Target recruitment is 62 patients from 14 participating UK centres. Patients with histological confirmation of invasive bladder cancer (T2-T4aN0M0) of any pathological

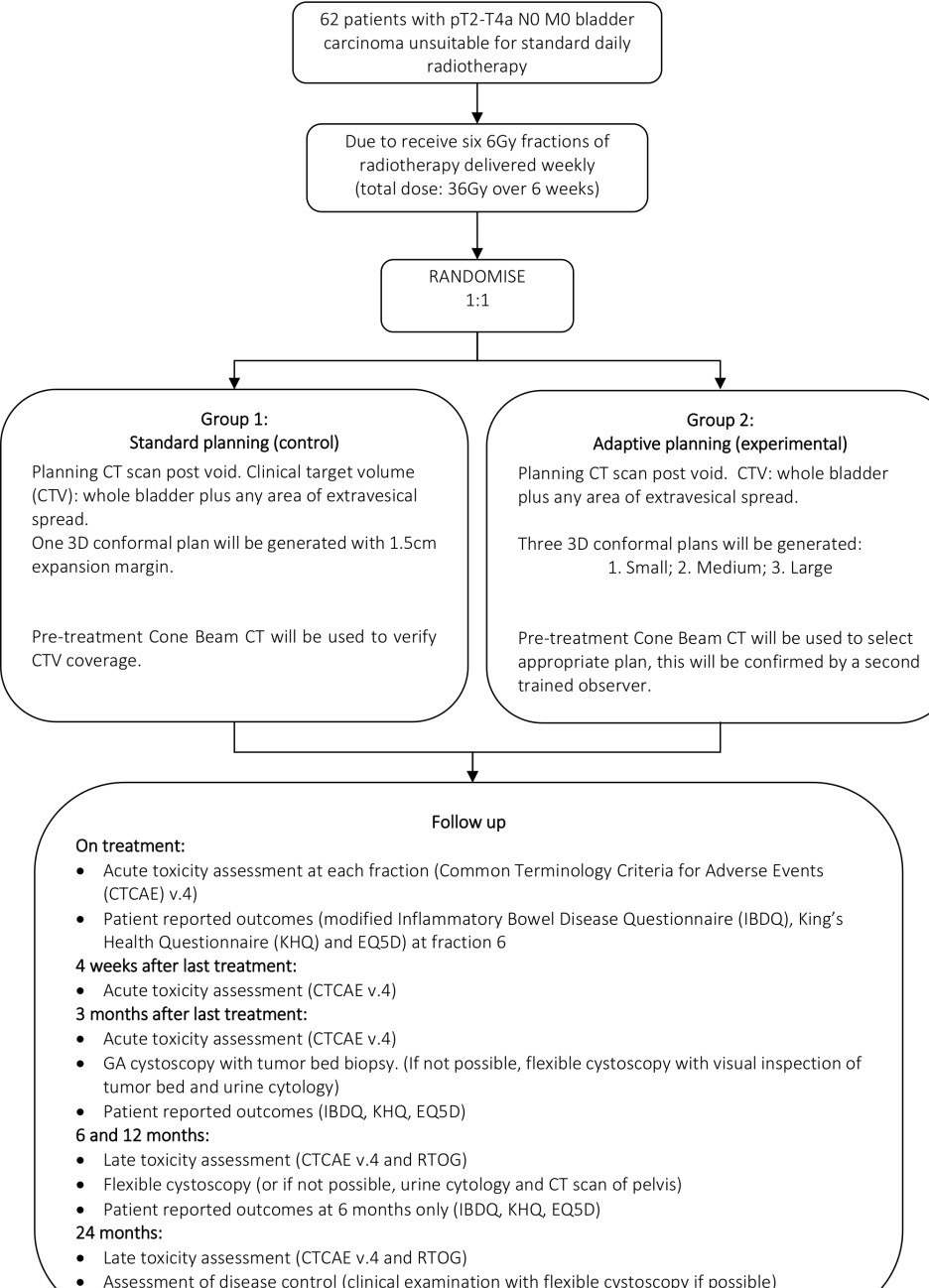

**Figure 1** Trial schema. RTOG, Radiation TherapyOncology Group.

subtype unsuitable for radical cystectomy or radical daily radiotherapy for any reason including but not limited to performance status, comorbidity or patient refusal will be approached for inclusion. Eligible patients would have an expected survival of greater than 6 months, be willing to accept assessment with cystoscopy following radiotherapy completion and be able to attend for follow-up.

Patients with an indwelling urinary catheter, active or history of other malignancy within 2 years of randomisation except for non-melanomatous skin carcinoma, previous non-muscle invasive bladder tumors and low-risk prostate cancer (as defined by National Comprehensive Cancer Network (NCCN) risk stratification as T1/T2a, Gleason 6 PSA <10) will be excluded. Those with history

of radiation to the pelvis or other contraindication to pelvic radiotherapy, for example, inflammatory bowel disease will also be excluded.

## Study treatment

All participants should have a transuretheral resection of the bladder tumor (TURBT) if possible prior to trial entry but this is not mandated, accepting that a proportion of patients will be unsuitable for this procedure. To permit sufficient time for radiotherapy planning, it is expected that treatment would commence within a maximum of 6 weeks from randomisation.

Participants will be planned to receive six, 6 Gy fractions delivered weekly to a total dose of 36 Gy. Those allocated

**Table 1** Schedule of assessments

| Visit/assessment | Prerandomisation | Up to 14 days pretreatment | On treatment (before each fraction) | 4 weeks after completion of radiotherapy | 3 months after completion of radiotherapy | 6 months after completion of radiotherapy | 12 months after completion of radiotherapy | 24 months after completion of radiotherapy | Annually thereafter |
|---|---|---|---|---|---|---|---|---|---|
| Histological confirmation of bladder cancer | X | | | | | | | | |
| Radiological assessment of bladder cancer (minimum CT abdomen and pelvis and chest X-ray) | X* | | | | | | | | |
| Acute toxicity assessment (CTCAE V.4) | | X | X | X | X | | | | |
| Full blood count, urea and electrolytes | | X | X† | | | | | | |
| Patient-reported outcomes questionnaire (IBDQ, KHQ and EQ5D) | | X | X‡ | | X | X | | | |
| Cystoscopy under general anaesthetic with tumor bed biopsy (if not possible, flexible cystoscopy with visual inspection of tumor bed and urine cytology) | | | | | X | | | | |
| Late toxicity assessment (CTCAE V.4 and RTOG) | | | | | | X | X | X | |
| Flexible cystoscopy with visual inspection of tumor bed (if not possible, urine cytology and pelvic CT scan) | | | | | | X | X | | |
| Assessment of disease status | | | | | | | | X | X |

*Baseline radiological assessment should take place ideally within 4 weeks and within a maximum of 6 weeks prior to randomisation.
†Full blood count, urea and electrolytes prior to fractions 2, 4 and 6 only.
‡PRO questionnaire at fraction 6 only.
CTCAE, Common Terminology Criteria for Adverse Events; IBDQ, Inflammatory Bowel Disease Questionnaire; KHQ, King's Health Questionnaire; RTOG, Radiation Therapy Oncology Group.

to the standard planning group will have one radiotherapy plan generated which will be used to deliver all six treatments. A CBCT scan acquired just prior to treatment delivery can be used to inform an online position correction in accordance with National Radiotherapy Implementation Group (NRIG) Report, on Image Guided Radiotherapy (IGRT)[20] and standard local practice.

Participants allocated to adaptive planning will have three radiotherapy plans generated corresponding to a small, medium and large PTV. A CBCT taken immediately prior to each treatment delivery will be used to select the most appropriate plan of the day depending on the bladder volume and shape. Plan selection is authorised to be carried out only by those radiographers or other practitioners (physicians or physicists) who have attained concordance with the gold standard PTV selection through the Radiotherapy Trials Quality Assurance (RTTQA) Group IGRT credentialing. This is to ensure all those participating in plan selection have the necessary advanced skill level required for the study.

### Radiotherapy planning and delivery
The radiotherapy planning and delivery guidance was developed in collaboration with the RTTQA Group.

### Radiotherapy planning CT scan
The patient preparation procedures are the same irrespective of randomisation arm. Patients are required to have an empty bladder for acquisition of the radiotherapy planning CT scan. Patients are therefore asked to void immediately before planning CT scan and not to drink fluids for 30 min before the planning scan. Given bladder deformation occurs with loaded rectum, patients are also encouraged to evacuate their bowels of flatus and faeces prior to scanning. The use of microenemas is permitted if it is standard local practice but is not mandated.

Patients are positioned supine with arms comfortably positioned out of the radiotherapy field using appropriate immobilisation devices. CT slices of ≤3 mm thickness are obtained from at least 4 cm above the dome of the bladder to 2 cm below the ischial tuberosities. No oral or intravenous contrast is required.

### Target volume definition
Volumes are defined according to the International Commission on Radiation Units and Measurements (ICRU) report 50, supplement report ICRU 62: Prescribing, Recording and Reporting Photon Beam Therapy and ICRU 83: Prescribing, Recording and Reporting Photon-Beam Intensity Modulated radiotherapy (IMRT).[21] Consistent structure naming convention for target volumes and organs at risk (OAR) is adopted for all patients participating within the trial.

Outlining should be carried out with the aid of all diagnostic MRI and CT scans wherever available. The clinical target volume (CTV) is contoured to encompass the gross tumor volume (GTV), the whole bladder and any area of extravesical spread. The CTV includes 1.5 cm of prostatic urethra in male patients or 1 cm of urethra in female patients if tumor is at the base of bladder or if distant carcinoma in situ (CIS) is present. It is not required that the GTV is drawn as a separate structure.

The CTV will be expanded either isotropically by 1.5 cm to create a single PTV for standard planning (control) or three PTVs using variable margins (small, medium and large) for adaptive planning depending on the randomisation arm. The CTV to PTV expansion details have been derived from earlier phase I/II work[17–19] and are summarised in table 2.

### Organs at risk delineation
OARs are identified as the rectum, other bowel and femoral heads. These structures are outlined as solid structures by defining their outer wall. The rectum is outlined to include the full circumference and rectal contents. The rectal outlining should extend from the lowest level of the ischial tuberosities to the rectosigmoid junction which identified as the level at which there is an anterior inflection of the bowel, best appreciated on sagittal reconstructions on the CT planning scan.

The small and large bowel (including sigmoid colon) is outlined as a single structure labelled other bowel. Small and large bowel visible on relevant axial slices of the planning scan is outlined as individual loops. The cranial extent of other bowel outlining should be 2 cm beyond the superior extent of the standard PTV or large PTV as appropriate.

Both the femoral heads are outlined to the bottom of the femoral head curvature. The femoral necks are not included.

**Table 2** Clinical target volume (CTV) to planning target volume (PTV) expansion details

| Patient randomisation | CTV to PTV expansion (cm) | | | | |
| --- | --- | --- | --- | --- | --- |
| | Laterally | Anteriorly | Posteriorly | Superiorly | Inferiorly |
| Standard plan | | | | | |
| PTV standard | 1.5 | 1.5 | 1.5 | 1.5 | 1.5 |
| Adaptive plan | | | | | |
| PTV small | 0.5 | 0.5 | 0.5 | 0.5 | 0.5 |
| PTV medium | 0.5 | 1.5 | 1 | 1.5 | 0.5 |
| PTV large | 0.8 | 2 | 1.2 | 2.5 | 0.8 |

## Radiotherapy planning

Three-dimensional conformal radiotherapy planning is recommended using three or four fields; however, use of static 5–7-field IMRT or volumetric modulated arc radiotherapy treatment is permitted. It is accepted that the preferred treatment planning method may vary between participating centres but should be stated at the start of the trial and then be used for all patients enrolled there.

For patient's randomised to standard planning, a single plan is created. For those patients randomised to adaptive planning, a series of three plans are created using PTV small, PTV medium and PTV large.

Three-dimensional dose distributions are produced for the overall prescribed dose of 36 Gy in six fractions. The dose distribution is assessed for coverage of the PTV and normal tissues sparing using appropriate transverse sagittal and coronal views.

All plans are created to ensure that at least 98% of the PTV (PTV $D_{98\%}$) receives >90% (ideally $\geq$95%) of the prescribed dose, the median PTV dose (PTV $D_{50\%}$) is within 1% of the prescription dose and the near maximum (PTV $D_{2\%}$) is $\leq$107% (ideally $\leq$105%) of the prescribed dose. To minimise unexpected high dose outside the PTV, it is required that 1 cm$^3$ of normal tissue outside the PTV should be $\leq$110% of the prescribed dose.

Dose to OARs should be as low as possible. To minimise dose to other bowel, it is recommended that the small plan for those randomised to adaptive radiotherapy aims to achieve the predefined optimal dose constraints, and the mandatory constraints for the medium plan. It is accepted that the rectum and bowel dose constraints of the large plan may not be met despite adequate optimisation. Assessment of other bowel dose on the large plan represents an overestimation of true dose to other bowel compared with when this plan is actually used to deliver treatment. This is because when the large plan is selected for treatment, a proportion of bowel moves out of the field with bladder filling. It is at the local principals' investigator discretion to accept the OARs doses.

**Table 3** Target volume constraints

| Dose constraints | Optimal | Mandatory |
|---|---|---|
| PTV $D_{98\%}$ | $\geq$95% of prescribed dose | $\geq$90% of prescribed dose |
| PTV $D_{50\%}$ | $\pm$1% of prescribed dose | – |
| PTV $D_{2\%}$ | $\leq$105% of prescribed dose | $\leq$107% of prescribed dose |
| Normal tissue $D_{1cc}$ | – | $\leq$110% of prescribed dose |

PTV $D_{98\%}$ is the dose received by 98% of planning target volume (PTV).
PTV $D_{50\%}$ is the dose received by 50% of PTV.
PTV $D_{2\%}$ is the dose received by 2% of PTV.
Normal tissue $D_{1cc}$ is the dose received by 1 cm$^3$ of normal tissue outside the PTV.

**Table 4** Organ at risk dose constraint guide

| Organ at risk | Constraint* | | |
| | Dose level | Optimal | Mandatory |
|---|---|---|---|
| Rectum | 17 Gy | 50% | 80% |
| | 28 Gy | 20% | 60% |
| | 33 Gy | 15% | 50% |
| | 36 Gy | 5% | 30% |
| Other bowel | V25 | 139 cm$^3$ | 208 cm$^3$ |
| | V28 | 122 cm$^3$ | 183 cm$^3$ |
| | V31 | 105 cm$^3$ | 157 cm$^3$ |
| | V33 | 84 cm$^3$ | 126 cm$^3$ |
| | V36 | 26 cm$^3$ | 39 cm$^3$ |
| Femoral heads | 28 Gy | – | 50% |

*The constraints provided serve only as a guide with recommendation that the optimal constraints particularly for other bowel should be met for the small plan and mandatory constraints should be met for medium plan.

The target volume and OAR dose volume constraints are summarised in tables 3 and 4, respectively.

### Preradiotherapy checks

To minimise risk of error at the time of plan importing, exporting and plan selection, it is recommended that each beam name and identification reflects the assigned plan. It is also important to ensure that the participating centre's local record and verify systems cannot mix beams from different plans at the time exporting from the treatment planning system and importing for treatment delivery. One way of achieving this is to create each plan with slightly different contributions from each field so that only the correct combination of beams can be chosen on any given day. Adding two points diagonally on the isocentre slice with a dose close to the 100% isodose would achieve this. All beams can then only be assigned from the same plan to each of the points as the reference point differs.

### Treatment delivery

The same patient preparation instructions used at planning CT will be implemented prior to each fraction delivered.

CBCT of the pelvis should be acquired prior to each fraction irrespective of randomisation. For those patients randomised to standard (control) arm, pretreatment CBCT should be used in accordance with guidance provided in the NRIG IGRT report.[20] It is therefore expected that this CBCT will inform appropriate corrections (either manual or automatic) to be applied prior to the delivered fraction to ensure that treatment is accurately directed.

For those patients randomised to the adaptive (experimental) arm, the pretreatment CBCT is acquired and registered to bone in accordance with the guidance provided in the NRIG IGRT report.[20] Appropriately

trained radiographers or other practitioners review the bone-matched CBCT assessing the bladder size and position in relation to the three PTVs and the coverage they provide. The PTV contour and corresponding plan providing the most suitable coverage with minimal normal tissue irradiation is selected. The most suitable contour is deemed to be that which encompasses the whole bladder CTV as seen on CBCT with an approximate 3 mm margin to account for any intrafraction filling that may occur during treatment delivery. A second appropriately trained radiographer or practitioner must confirm the selected PTV and corresponding plan. Once agreement has been reached, any necessary couch correction is performed prior to treatment delivery with the selected plan.

If no PTV contour appears to provide suitable coverage of the bladder CTV, then it is advised that the patient is removed from the treatment couch and is asked to empty their bladder and, or bowel. The above steps are repeated with CBCT acquired just prior to treatment to reassess bladder. It advised that the centre contacts the RTTQA Group for advice if the PTV still appears to provide inadequate target coverage.

### Treatment scheduling

Treatment can be scheduled to start on any day of the week but each fraction should be delivered on the same day of the week at weekly intervals±2 days. Therefore, a maximum interval of 9 days between fractions is acceptable in the event of machine breakdown or service. For any gaps longer than this, the participating centre is advised to contact the trial team.

### Radiotherapy protocol compliance programme

A comprehensive radiotherapy quality assurance (QA) programme led by the RTTQA Group has been implemented for the HYBRID trial, and has been previously described.[22 23] The QA programme aims to standardise contouring, planning and delivery of image guided and adaptive bladder radiotherapy in participating centres. It comprises of both pretrial and on-trial components including independent monitoring of appropriate treatment plan selection for the adaptive planning during patient recruitment.

Prior to trial, entry participating centres are asked to complete an online facility questionnaire in order to gauge current local IGRT experience. A separate process document is used to collect task details of all aspects of a complete patient pathway.

The principal investigator (PI) at each participating centre is asked to contour two benchmark clinical cases as per protocol. Structured feedback is provided via RTTQA Group to the PI.

All participating trial centres are required to complete a planning benchmark case. Centres are provided with access to CT Digital Imaging and Communications in Medicine (DICOM) data and preoutlined structure set. They are requested to then plan this patient in their own treatment planning system as if randomised to the HYBRID adaptive arm. It is the responsibility of the local investigator to ensure that appropriate plan checking QA process is in place at their local institution. Once the three plans of the benchmark case have been created, reviewed and accepted by the local PI, the DICOM CT, dose cubes, RT plan and structure sets are returned in to the RTTQA Group via secure file transfer and structured feedback is provided.

It is a pretrial requirement that all participating centres have both an established IGRT training programme in place for their radiographers and be using CBCT to assess bladder treatment delivery. Trial-specific bladder IGRT competency is completed through an online training package, practical workshop and independent assessment of plan selection.

The online training consists of three practice cases each with six CBCTs to work through. Step-by-step instructions with correct plan selections is provided. Following this, a credentialing assessment consisting of 12 plan selections is carried out. The plan selections and matched reviews are assessed by the RTTQA Group and structured feedback provided. Only those who meet minimum threshold of concordance of plan selection as predefined by the trial team will be approved for performing HYBRID plan selection.

As part of the on-trial QA, each participating centre visited by the RTTQA Group during their first adaptive patient's treatment course for an on-site review of the local image registration processes and plan selection decision-making. Once the first adaptive patient has been recruited from each participating centre, the plans and plan selections for treatment delivery will be retrospectively reviewed remotely prior to the second patient starting treatment.

All planning data and treatment delivery data (CBCT, registration objects and treatment forms) are collected and reviewed by the RTTQA Group to ensure adherence to the HYBRID planning and delivery protocol is maintained. Remote retrospective plan selection review will take place for all adaptive radiotherapy patients during the trial.

### Statistical considerations

The primary objective is to assess whether adaptive radiotherapy techniques when delivered at multiple centres can lead to a reduction in the level of acute non-GU toxicity experienced by patients with MIBC unsuitable for daily radical radiotherapy.

The sample size is based on the primary endpoint of acute (up to 3 months after the end of radiotherapy) non-GU CTCAE ≥grade 3 toxicity. An A'Hern exact phase II design was used to rule out an upper limit for each planning method separately. Based on results of the non-randomised single centre phase II feasibility study of adaptive predictive planning for hypofractionated bladder radiotherapy (APPLY study; NCT01000129),[18] it is expected that the acute non-GU ≥grade 3 rate will be 10% (p1=0.9) in patients receiving adaptive planning. The

study is designed to rule out a 30% (p0=0.7) upper limit of ≥grade 3 non-GU toxicity with each planning method. For 80% power (β=0.2) and 5% alpha (one-sided) in each planning group, 28 evaluable patients are required and if 5 or more experience non-GU ≥grade 3 toxicity, then the acute toxicity associated with that planning technique will be assumed to be too high. To be evaluable for acute toxicity, participants must receive at least one fraction of radiotherapy. Incorporating a 10% non-evaluable rate gives a target sample size of 62 patients (31 in each planning group).

The numbers and proportions of patients with acute non-GU CTCAE V.4 toxicity ≥grade 3 within the first 3 months of completing radiotherapy in each planning method will be presented together with 95% one-sided exact CIs (the 90% two-sided CI will also be presented).

Late toxicity will be summarised by frequencies and proportions at each time point by treatment group. Kaplan-Meier methods will be used to present time to event outcomes; due to small numbers, no formal comparison is planned.

### Ethics
The trial is approved by the London-Surrey Borders Research Ethics Committee (13/LO/1350).

### Safety reporting
Data are collected at each trial visit regarding any adverse events according to the CTCAE V.4.0 grading system. The highest grade observed since the last visit should be reported. All serious adverse events (SAEs) are reported to the ICR-CTSU within 24 hours of the PI becoming aware of the event. SAEs should be followed up until clinical recovery is complete or until the condition has stabilised. Any safety concerns will be reported to the main research and ethics committee by ICR-CTSU as part of the annual progress report.

### Trial monitoring and oversight
The trial is supervised by a Trial Management Group (TMG) that includes the Chief Investigator, trials unit scientific lead, statistician and coordinators along with coinvestigators, identified collaborators including RTTQA Group representative, and lay/consumer representative.

Oversight is provided by an independent trials steering committee and an independent data monitoring committee (IDMC).

There are no formal early stopping rules for efficacy or toxicity but, as per the statistical design, if five or more participants report non-GU ≥grade 3 toxicities in one planning group then randomisation will cease. The IDMC would then review the data and advise on continuation of recruitment to the other planning method.

### Trial status and dissemination of results
The first patient was registered in April 2014. The study completed recruitment in August 2016. It is expected that the trial will report in 2020. The results will be disseminated via peer-reviewed scientific journals, conference presentations and submission to regulatory authorities.

### Patient and public involvement
The HYBRID trial has been reviewed and endorsed by patient and carer representatives from the National Cancer Research Institute (NCRI) Consumer Liaison Group and the NCRI Clinical and Translational Radiotherapy Research Group working group.

Patient and public involvement began at the protocol design and development stage via national and local consumer oversight committee review. This included the NIHR Biomedical Research Centre radiotherapy studies consumer panel at the ICR and The Royal Marsden NHS Foundation Trust, and the NCRI Bladder Clinical Studies Group, which includes consumer representation.

Patients who had participated in the phase I study were asked to assess the burden of involvement required for participation in the HYBRID trial. This included review of the patient-reported outcomes questionnaires.

The trial patient information sheet and consent form were reviewed by the South West London Cancer Research Network consumer group. Their feedback was adopted and incorporated in to the final version of both documents. Copy of the ethics approved final version of the patient information sheet and consent is provided in the online supplementary file 2.

Patient representation on the TMG advises on day-to-day management of the trial including patient recruitment, and it is expected that they will also participate in dissemination of results via patient groups with bladder cancer.

**Author affiliations**
[1]Radiotherapy and Imaging, The Institute of Cancer Research, London, UK
[2]Radiotherapy and Imaging, The Royal Marsden Hospital NHS Trust, London, UK
[3]Mount Vernon Hospital, National Radiotherapy Trials Quality Assurance Group, Northwood, UK
[4]Laboratory of Radiation Physics, Odense University Hospital, Odense, Denmark
[5]Clinical Trials and Statistics Unit, The Institute of Cancer Research, London, UK

**Acknowledgements** SH, KW-O, HM, RL, EH and RH acknowledging this study represents independent research supported by the National Institute for Health Research (NIHR) Biomedical Research Centre at The Royal Marsden NHS Foundation Trust and The Institute of Cancer Research, London.

**Contributors** All contributors meet at least of one the criteria recommended by the ICMJE. RH and EH conceived the study design. SH, KW-O, HM, VH, RL, EH and RH were involved in protocol development. SH wrote the first draft of the radiotherapy protocol and manuscript. All authors contributed to subsequent drafts and revisions of the radiotherapy protocol and manuscript.

**Funding** The HYBRID trial is funded by Cancer Research UK (CRUK/12/055). Cancer Research UK supports the work of the Clinical Trials and Statistics Unit at The Institute of Cancer Research (ICR-CTSU) through a programme grant award (1491/A15955). The trial was supported by the National Radiotherapy Trials Quality Assurance Group (RTTQA).

**Disclaimer** The views expressed are those of the author(s) and not necessarily those of the NIHR or the Department of Health and Social Care.

**Competing interests** SH reports non-financial support from Elekta (Elekta AB, Stockholm, Sweden), non-financial support from Merck Sharp & Dohme, personal fees and non-financial support from Roche outside the submitted work; EP, AW, KW-O, VH, HM, EM, and RL have no conflicts to disclose; EH reports grants from

Cancer Research UK during the conduct of the study; grants from Accuray, grants from Varian Medical Systems, outside the submitted work; RH reports non-financial support from Janssen, grants and personal fees from MSD, personal fees from Bristol Myers Squibb, grants from CRUK, other from Nektar, personal fees and non-financial support from Roche, outside the submitted work.

**Patient and public involvement** Patients and/or the public were involved in the design, or conduct, or reporting, or dissemination plans of this research. Refer to the Methods section for further details.

**Patient consent for publication** Not required.

**Provenance and peer review** Not commissioned; externally peer reviewed.

**ORCID iD**
Shaista Hafeez http://orcid.org/0000-0002-2057-0946

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
