## [Reviewer comments · BMJ Open]

ARTICLE DETAILS

TITLE (PROVISIONAL)	Protocol for hypofractionated adaptive radiotherapy to the bladder within a multi-centre phase II randomised trial: radiotherapy planning and delivery guidance
AUTHORS	Hafeez, Shaista; Patel, Emma; Webster, Amanda; Warren-oseni, Karole; Hansen, Vibeke; McNair, Helen; Miles, Elizabeth; Lewis, Rebecca; Hall, Emma; Huddart, Robert

VERSION 1 - REVIEW

REVIEWER	Stefano Arcangeli University of Milan Bicocca, Italy
REVIEW RETURNED	13-Feb-2020

GENERAL COMMENTS	In this protocol the authors aim to randomly compare single standard plan or adaptive radiotherapy using a library of three plans (small, medium, and large) in the hypofractionated treatment of T2-T4N0M0 bladder cancer patients unsuitable for cystectomy or normofractionated daily radiotherapy. The study design is appropriate and answer an unmet clinical need, since plan of the day strategy holds promises in minimizing the dose to nearby normal tissues, and namely the intestine. As such, the choice of genitourinary grade 3 or greater toxicity as primary endpoint seems robust enough to support this hypothesis. If proven true, such an approach will be probably implemented in the daily clinical practice and inform clinicians on the tolerance of hypofractionated radiotherapy in this setting. There are however some minor issues that I'd like to point out: 1) The strategy herein proposed rely on the assumption that patient's set up and anatomy do not vary from week to week and that the intrafractional motion is negligible. It is likely that in the real practice the position of some organs at risk, like the bowel, may significantly vary.2) The online image registration process needed to accomplish the plan of the day usually takes a relevant amount of extra-time, which may result in the risk of uncertainties in patients' compliance and setup, and turns to be potentially detrimental. The authors are invited to provide their comments on these criticisms
---

REVIEWER	Ramses Sadek Augusta University, USA
REVIEW RETURNED	18-Feb-2020

GENERAL COMMENTS	The title is a little confusing. The Phase II study usually assess the preliminary efficacy while this study focuses on toxicity. The sample size justification are not clear. I would like to see clear primary objective defined in the stat section along with analysis methods that drive the sample size and power estimation. In the abstract it indicated that the $p_0=0.71$, $p_1=0.90$ for the given probability of Type I and Type II errors. There is no mention of these proportions in the statistical considerations or what do they mean.
--

VERSION 1 – AUTHOR RESPONSE

Reviewer: 1

Reviewer Name: Stefano Arcangeli

Institution and Country: University of Milan Bicocca, Italy

Please state any competing interests or state 'None declared': I declare no conflicts of interest

In this protocol the authors aim to randomly compare single standard plan or adaptive radiotherapy using a library of three plans (small, medium, and large) in the hypofractionated treatment of T2-T4N0M0 bladder cancer patients unsuitable for cystectomy or normofractionated daily radiotherapy. The study design is appropriate and answer an unmet clinical need, since plan of the day strategy holds promises in minimizing the dose to nearby normal tissues, and namely the intestine. As such, the choice of genitourinary grade 3 or greater toxicity as primary endpoint seems robust enough to support this hypothesis. If proven true, such an approach will be probably implemented in the daily clinical practice and inform clinicians on the tolerance of hypofractionated radiotherapy in this setting.

We thank the reviewer for his positive comments, however it should be noted that the study is not powered for formal comparisons of standard versus adaptive radiotherapy and that the primary endpoint is acute non-genitourinary toxicity not genitourinary toxicity as stated.

There are however some minor issues that I'd like to point out:

1) The strategy herein proposed rely on the assumption that patient's set up and anatomy do not vary from week to week and that the intrafractional motion is negligible. It is likely that in the real practice the position of some organs at risk, like the bowel, may significantly vary.

The adaptive plan of the day/library of plans strategy is employed because we acknowledge significant anatomical variation between (inter-fraction) and during (intra-fraction) radiotherapy exists (please see p 5 para 4). We and others have previously published on the magnitude of the anatomical changes occurring during bladder radiotherapy. This work is referenced within the manuscript and can be found in references 14-16 and 19.

We also acknowledge the significance of intra-fraction filling. To mitigate against this we recommend that

'the most suitable contour to inform selection is deemed to be that which encompasses the whole bladder CTV as seen on CBCT with an approximate 3mm margin to account for any intra-fraction filling that may occur during treatment delivery' (please see p12 para 2).

Target coverage as assessed on the post treatment CBCTs is accepted measure of whether intra-fraction anatomical changes have been appropriately accounted for. In our phase I one work (reference 19) which informed the adaptive technique outlined in this manuscript, the mean bladder target (CTV) coverage as assessed on the post treatment CBCT V95 was 99% of prescription dose (standard deviation 2.0; range 91-100%).

No further changes made to the manuscript.

2) The online image registration process needed to accomplish the plan of the day usually takes a relevant amount of extra-time, which may result in the risk of uncertainties in patients' compliance and setup, and turns to be potentially detrimental.

We recognise the importance of additional time potentially having a detrimental effect on target coverage. A surrogate measure of the time take for the plan of the day workflow is to determine the mean time between the pre- and post-treatment CBCT scans. This time frame captures the image assessment, plan selection, couch correction and patient treatment. In our phase I work mean time between pre and post treatment CBCT was 14 minutes (reference 19). As described above, this time did not have a detrimental effect on target coverage on the post treatment CBCT by the selected plan. In order to ensure that approach can be successfully translated into the multi-centre setting we describe the training package required to be completed prior to trial participation (see p13 para 5 and 6). Each participating centre is also visited by the National Radiotherapy Trials Quality Assurance (RTTQA) group during their first adaptive patient's treatment course for an on site review of the local image registration processes and plan selection decision-making is both accurate and can be performed within a reasonable time frame (see p14 para 1).

No further changes made to the manuscript

Reviewer: 2

Reviewer Name: Ramses Sadek

Institution and Country: Augusta University, USA

Please state any competing interests or state 'None declared': None declared

The title is a little confusing. The Phase II study usually assess the preliminary efficacy while this study focuses on toxicity.

The primary goal of the intervention in this clinical setting is to improve the therapeutic risk: benefit ratio through toxicity reduction and not improvement in cancer control. Phase II trials of technical innovation are conducted to provide evidence of early clinical effectiveness and safety of innovation. It is widely accepted that effectiveness can be measured in terms of toxicity in radiation oncology clinical trials, particularly in the phase II setting.

Further details of the framework for systemic clinical evaluation in radiation oncology can be found here, Verkooijen HM et al., R-IDEAL: A Framework for Systematic Clinical Evaluation of Technical Innovations in Radiation Oncology. *Frontiers in oncology* 2017, 7:59.

The sample size justification are not clear. I would like to see clear primary objective defined in the stat section along with analysis methods that drive the sample size and power estimation. In the abstract it indicated that the $p_0=0.71$, $p_1=0.90$ for the given probability of Type I and Type II errors. There is no mention of these proportions in the statistical considerations or what do they mean.

The following sentence has been added to the statistical considerations section to reiterate the primary objective:

The primary objective is to assess whether adaptive radiotherapy techniques when delivered at multiple centres can lead to a reduction in the level of acute non-genitourinary (GU) toxicity experienced by patients with muscle invasive bladder cancer unsuitable for daily radical radiotherapy.

The statement in the abstract has been corrected to reflect $p_0=0.70$, $p_1=0.9$, $\alpha=0.05$, $\beta=0.2$, and this been added to the descriptive sample size justification (p14 para 4) which has been amended for clarity as follows:

The sample size is based on the primary endpoint of acute (up to 3 months after the end of radiotherapy) non-genitourinary CTCAE >grade 3 toxicity. An A'Hern exact phase II design was used to rule out an upper limit for each planning method separately. Based on results of the APPLY study (NCT01000129) [18], it is expected that the acute non-genitourinary >grade 3 rate will be 10% ($p_1=0.9$) in patients receiving adaptive planning. The study is designed to rule out a 30% ($p_0=0.7$) upper limit of >grade 3 non-genitourinary toxicity with each planning method. For 80% power ($\beta=0.2$) and 5% alpha (one-sided) in each planning group, 28 evaluable patients are required and if 5 or more experience non-genitourinary >grade 3 toxicity then the acute toxicity associated with that planning technique will be assumed to be too high. To be evaluable for acute toxicity participants must receive at least 1 fraction of radiotherapy. Incorporating a 10% non-evaluable rate gives a target sample size of 62 patients (31 in each planning group).

The descriptive summary statistics used to analyse the primary endpoint are described on p14 para 5

VERSION 2 – REVIEW

REVIEWER	Stefano Arcangeli University of Milan Bicocca S. Gerardo Hospital 20900-Monza (Italy)
REVIEW RETURNED	31-Mar-2020

GENERAL COMMENTS	The authors have appropriately addressed all comments raised by the reviewers and the manuscript is suitable for publication.
---